# Ethylene electrosynthesis from low-concentrated acetylene via concave-surface enriched reactant and improved mass transfer

Fanpeng Chen [1,3], Li Li [2,3], Chuanqi Cheng [1,3], Yifu Yu [2], Bo-Hang Zhao [1,2] ✉ & Bin Zhang [1] ✉

Electrocatalytic semihydrogenation of acetylene ($C_2H_2$) provides a facile and petroleum-independent strategy for ethylene ($C_2H_4$) production. However, the reliance on the preseparation and concentration of raw coal-derived $C_2H_2$ hinders its economic potential. Here, a concave surface is predicted to be beneficial for enriching $C_2H_2$ and optimizing its mass transfer kinetics, thus leading to a high partial pressure of $C_2H_2$ around active sites for the direct conversion of raw coal-derived $C_2H_2$. Then, a porous concave carbon-supported Cu nanoparticle (Cu-PCC) electrode is designed to enrich the $C_2H_2$ gas around the Cu sites. As a result, the as-prepared electrode enables a 91.7% $C_2H_4$ Faradaic efficiency and a 56.31% $C_2H_2$ single-pass conversion under a simulated raw coal-derived $C_2H_2$ atmosphere (~15%) at a partial current density of 0.42 A cm$^{-2}$, greatly outperforming its counterpart without concave surface supports. The strengthened intermolecular $\pi$ conjugation caused by the increased $C_2H_2$ coverage is revealed to result in the delocalization of $\pi$ electrons in $C_2H_2$, consequently promoting $C_2H_2$ activation, suppressing hydrogen evolution competition and enhancing $C_2H_4$ selectivity.

The production of the essential chemical ethylene ($C_2H_4$) is highly dependent on high-temperature naphtha cracking, which relies on petroleum resources with excess carbon emissions[1,2]. Hence, developing a petroleum-independent and mild strategy for $C_2H_4$ production is highly desirable for a low-carbon economy[3–8]. Recently, the electrocatalytic semihydrogenation of coal-derived acetylene ($C_2H_2$) to ethylene (ESAE) strategy has been developed[7,8]. Inhibiting the competing hydrogen evolution reaction (HER) at an industrial current density (≥200 mA cm$^{-2}$) is pivotal for the economic potential of the ESAE strategy. At present, both the $C_2H_4$ Faradaic efficiency (FE) and the optimal current density are extremely low for low-concentration $C_2H_2$ hydrogenation (e.g., <50% FE at 60 mA cm$^{-2}$ for ~1% $C_2H_2$ impurity hydrogenation in $C_2H_4$), which is far from the target of practical $C_2H_4$

production[7]. Additionally, the cost of separating and concentrating $C_2H_2$ feed gas accounts for a large proportion of the total $C_2H_4$ production cost[8]. Consequently, the cost of $C_2H_4$ production would further decrease if the raw tail gas (~15% $C_2H_2$) from the arc-plasma process of coal could be directly used as feedstock for the ESAE process[8–13]. However, the HER dominates the whole process as the $C_2H_2$ concentration decreases (Fig. 1a). Therefore, further development of highly efficient and selective catalysts for converting raw coal-derived $C_2H_2$ into $C_2H_4$ with high selectivity and conversion rates is urgently needed.

Generally, for a gas-involved reaction, enriching its local concentration and boosting the mass transfer toward active sites are important for enhancing the reactant's partial pressure to improve the

[1]Department of Chemistry, School of Science, Tianjin University, Tianjin 300072, China. [2]Institute of Molecular Plus, Tianjin University, Tianjin 300072, China. [3]These authors contributed equally: Fanpeng Chen, Li Li, Chuanqi Cheng. ✉e-mail: bhzhao@tju.edu.cn; bzhang@tju.edu.cn

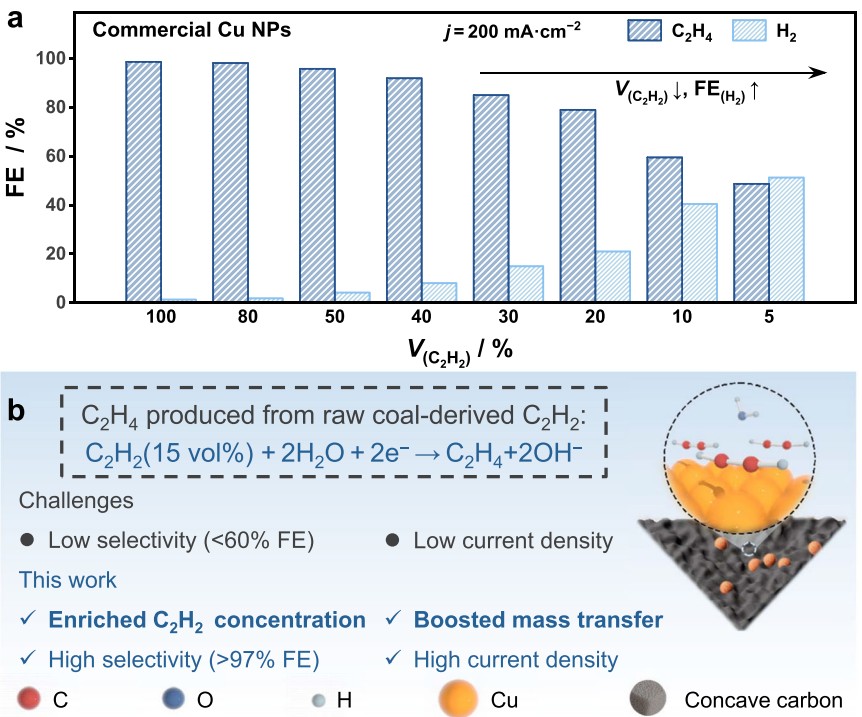

**Fig. 1 | Reasons and principles for the design of Cu-PCC catalysts. a** The performance of the ESAE under different $C_2H_2$ concentrations at a current density of 200 mA cm$^{-2}$ over commercial Cu nanoparticles. **b** Illustration of the principles of our proposed strategy.

activity and selectivity[14–17]. Copper (Cu) nanoparticles have been proven to be an appropriate choice for suppressing the competing HER under high $C_2H_2$ partial pressure[18–21]. Thus, the critical issue for the selective conversion of raw coal-derived $C_2H_2$ lies in enriching the concentration and guaranteeing the facile mass transfer of $C_2H_2$ around the surface of the Cu nanoparticles. Structures with high curvatures always lead to a high local electric field, which can gather reactants around the catalyst surface and increase its concentration[22–26]. For example, Liu et al. demonstrated that Cu nanoneedles could increase the adsorption of the *CO intermediate and, in turn, accelerate C−C coupling during the electrocatalytic $CO_2$ reduction process[23]. In addition to the tips, the concave surface also has a high curvature[27–29]. Moreover, carbon-based supports with porous structures could effectively boost gas capture and transport, benefitting mass transfer[30–32]. In this regard, Cu nanoparticles loaded on porous carbon supports with abundant nanosized concave surfaces (denoted as Cu-PCC) are expected to be efficient at increasing the concentration and increasing the mass transfer of $C_2H_2$ around Cu sites through the unique concave support, consequently suppressing the HER under low-concentration raw coal-derived $C_2H_2$ (Fig. 1b). However, the synthesis and exploration of porous concave carbon-supported Cu nanoparticle electrodes for electrocatalytic $C_2H_4$ production are lacking.

Herein, a preliminary density functional theory (DFT) calculation was first conducted to show that a concave surface is beneficial for the enrichment and facile mass transfer of $C_2H_2$, increasing its partial pressure around the active sites. Then, we designed a facile self-template method to synthesize porous concave carbon-supported Cu nanoparticle (Cu-PCC), which was found to be an outstanding electrocatalyst for the ESAE process, using simulated raw coal-derived $C_2H_2$ as feedstocks. Cu-PCC delivered a $C_2H_4$ FE of 91.70% and a single-pass $C_2H_2$ conversion of 56.31% at a potential of −1.2 V versus a reversible hydrogen electrode (vs. RHE) at a partial current density of 0.42 A cm$^{-2}$, greatly outperforming the Cu nanoparticles supported on carbon without a concave surface counterpart. Moreover, $C_2H_2$ temperature-programmed desorption ($C_2H_2$-TPD) and in situ

spectroscopic characterization experiments revealed that the polarization field induced by the concave surface over Cu-PCC increased $C_2H_2$ coverage and strengthened the intermolecular π-conjugation of $C_2H_2$, thus leading to the delocalization of the π electrons of $C_2H_2$ to promote the activation of $C_2H_2$ and enhance the $C_2H_4$ selectivity of the ESAE with raw coal-derived $C_2H_2$.

## Results
### The design and synthesis of an electrocatalyst
We first conducted DFT calculations to evaluate the local field induced by the concave surface. As shown in Fig. 2a, the electrons are enriched at the concave carbon surfaces to build a polarization field, which could enhance the conjugation between $C_2H_2$ and the negative center (Fig. 2b and Supplementary Fig. 1), thus leading to the downshifting of the bonding orbital and benefiting the enrichment of $C_2H_2$ (Fig. 2c)[21,29,33]. Once low-concentration $C_2H_2$ accumulated on the concave carbon surfaces, facile migration to the Cu sites was still a prerequisite for the following reaction. In that case, simulations of the migration pathway of the $C_2H_2$ molecule in solution over the Cu-C and Cu-PCC interfaces were conducted (Supplementary Figs. 2–3). For gas-involved reactions, there will be a few layers of water clusters (WC) due to the hydrogen bonding network around the gas–solid–liquid three-phase interface, and the gap between the WC and solid surface (labeled *d* in Fig. 2d) provides a diffusion channel for gaseous reactants. As shown in Fig. 2d, the diffusion channel for Cu-PCC is larger than that for Cu-C, thus leading to a straight-line migration pathway rather than a distorted pattern over the counterpart. To further quantify *d* over the C and PCC models, the radial distribution functions (RDFs) between C −H were calculated. As shown in Supplementary Fig. 4, $g(r)_{C-H}$, which is closely related to *d*, shows an average increase over the PCC model at approximately 0.1 Å compared to its C counterpart. This result indicates that the *d* over the concave C layer is lengthened. In addition, the associated migration energy barriers were calculated, as shown in Fig. 2e. The maximum migration energy for $C_2H_2$ diffusion over Cu-PCC is 0.41 eV, which is much lower than that of Cu−C (1.23 eV),

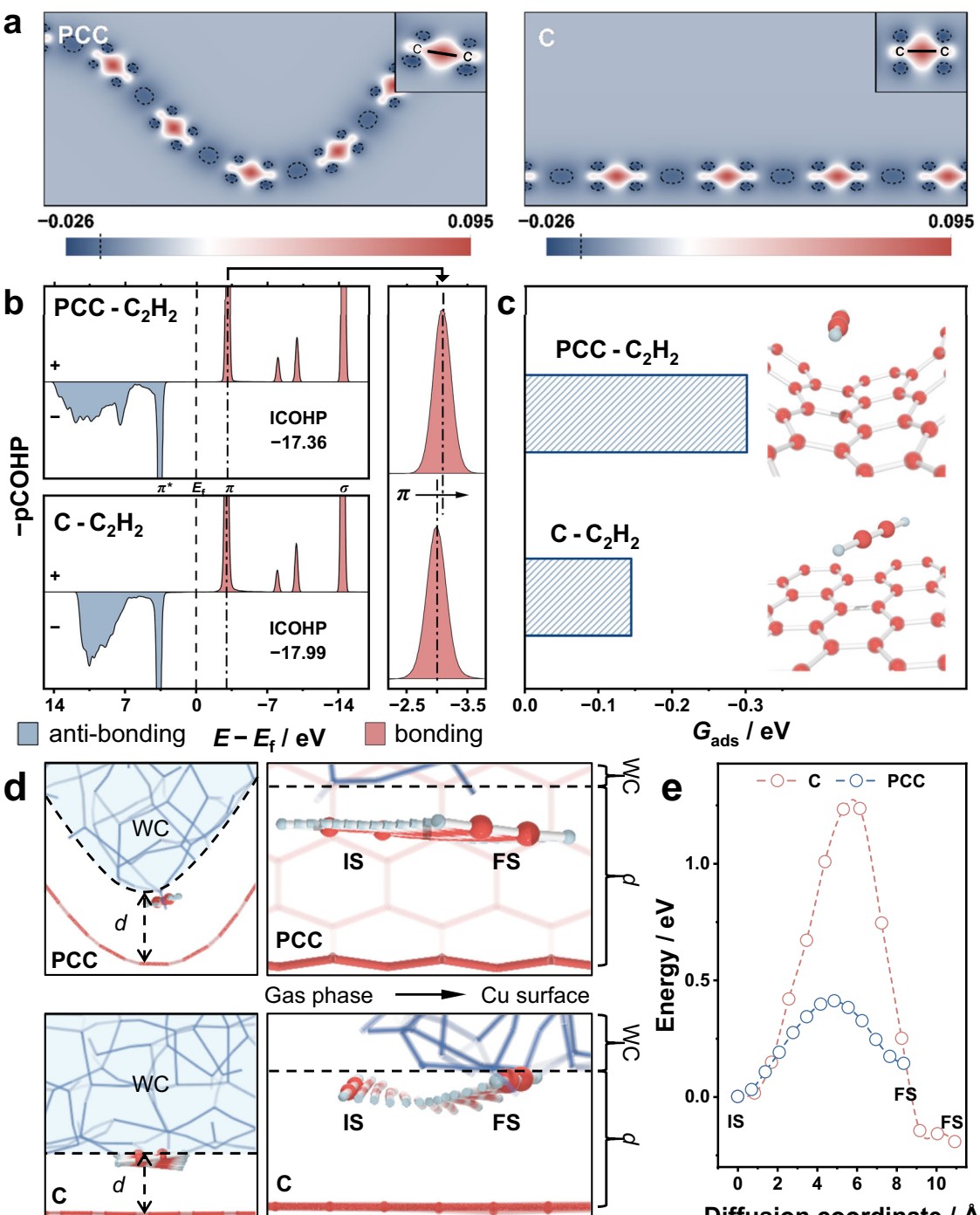

**Fig. 2 | Theoretical prediction of the C$_2$H$_2$ enrichment ability of a carbon support with concave surfaces. a** Projected crystal orbital Hamilton population (−pCOHP) for the C–C interaction of C$_2$H$_2$. **b** −pCOHP for the C–C interaction of C$_2$H$_2$ on PCC and C. **c** Adsorption of C$_2$H$_2$ on PCC and C. **d** C$_2$H$_2$ migration pathway illustration. **e** The energy barriers of Cu-PCC and Cu-C.

indicating that the mass transfer of C$_2$H$_2$ is significantly greater over the concave surface. These theoretical results indicate that a carbon support with concave surfaces could efficiently gather low-concentration raw coal-derived C$_2$H$_2$ feedstocks and increase the mass transfer kinetics for subsequent hydrogenation over Cu sites.

Generally, the collapse and reconstruction of a surface increases the roughness and results in many nanosized concave surfaces[34,35]. Thus, a sequential self-template transformation method based on the Kirkendall effect was proposed for the synthesis of Cu-PCC (Fig. 3a)[36,37]. Scanning electron microscopy (SEM) and transmission electron

microscopy (TEM) confirmed the successful preparation of Cu-based metal-organic framework precursors (Cu-MOF) with planar surfaces and octahedral-like morphologies (Supplementary Fig. 5). After the reaction of Cu-MOF precursors with tannic acid (TA) under optimum conditions, the octahedron-like shapes can be maintained, and the surface collapses inwards (denoted as Cu-TA, Supplementary Figs. 6, 7). After the annealing of Cu-TA under an H$_2$ atmosphere, the wall of the Cu-TA complex was converted to porous carbon with abundant nanosized concave surfaces, as confirmed by scanning transmission electron microscopy (STEM), SEM, and TEM images (Fig. 3b, c and

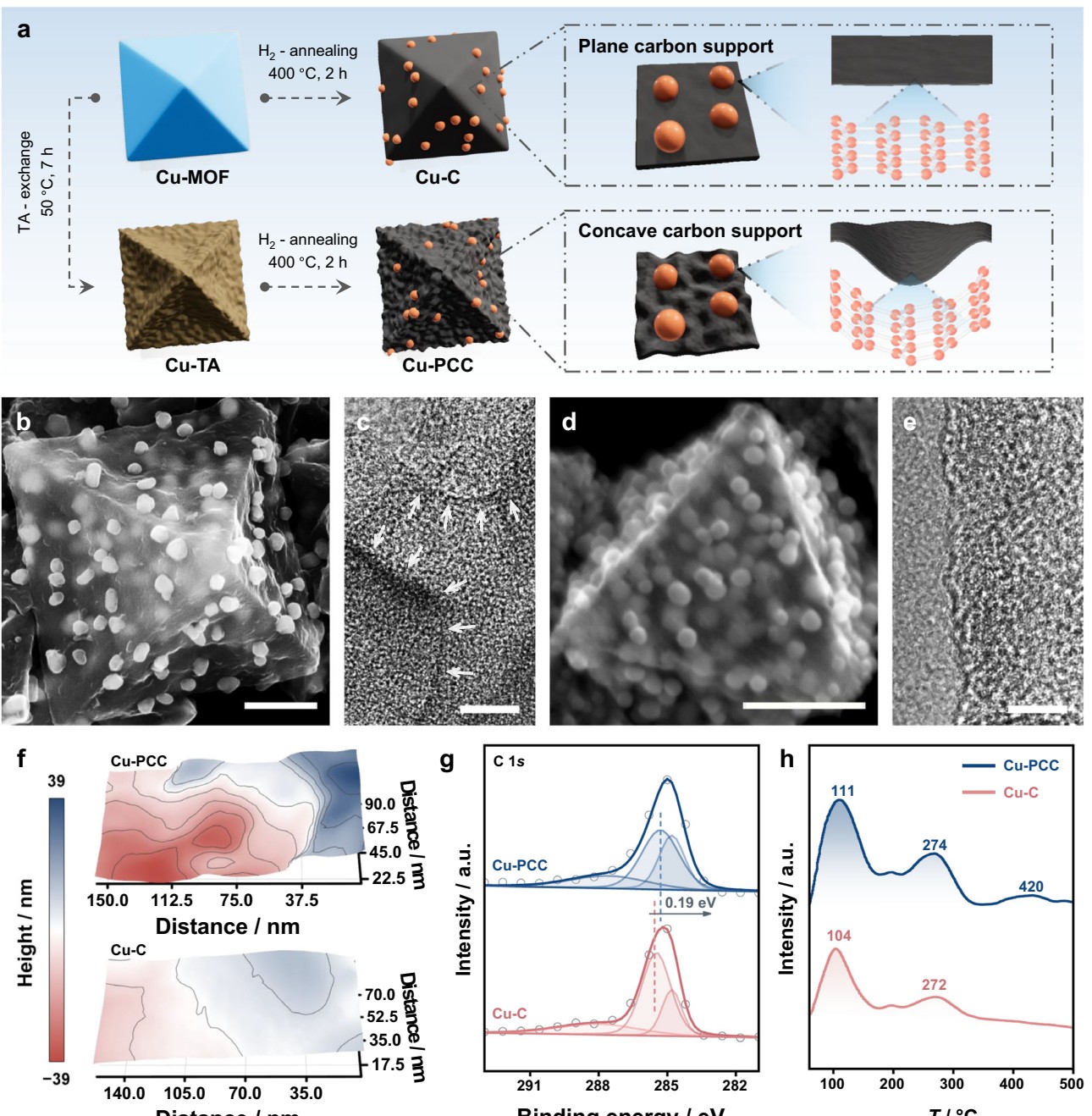

**Fig. 3 | Synthesis and Characterization of Cu-PCC and Cu-C. a** Schematic diagram illustrating the synthetic process of Cu-PCC. **b–e** STEM and corresponding TEM images of Cu-PCC (**b**, **c**) and Cu-C (**d**, **e**). STEM scale bar, 400 nm; TEM scale bar, 10 nm. **f** The height distributions of Cu-PCC and Cu-C obtained from the AFM images. **g** C 1$s$ XPS spectra of Cu-PCC and Cu-C. **h** $C_2H_2$-TPD of Cu-PCC and Cu-C.

Supplementary Fig. 8)[38]. However, Cu-C directly calcinated from Cu-MOF precursors without the collapse process exhibited a planar carbon surface (Fig. 3d, e and Supplementary Fig. 9). In addition, the size distributions were determined from three STEM images of Cu-PCC and Cu-C. As shown in Supplementary Fig. 10, the size of the majority of Cu particles in Cu-PCC is approximately 80 nm, which is slightly larger than that over Cu-C (~60 nm). Moreover, the atomic force microscopy (AFM) images also demonstrated the rougher surface of Cu-PCC caused by these concave surfaces compared to that of Cu-C (Fig. 3f and Supplementary Fig. 11). In addition, X-ray diffraction (XRD) patterns, Fourier transform infrared (FTIR) spectra, and Raman spectra were obtained to monitor the sequential conversion process from Cu-MOF

precursors to Cu-TA and eventually to Cu-PCC (Supplementary Fig. 12). The Raman, X-ray photoelectron spectroscopy (XPS), X-ray absorption spectroscopy (XAS), and contact angle results show that there are no other differences between Cu-PCC and Cu-C, other than the nanosized concave surfaces over the PCC supports (Supplementary Figs. 13–16). Note that the negative shift in the binding energies of C−O and C=O over Cu-PCC compared to that over Cu-C verifies the existence of a polarization field (Fig. 3g)[39–41]. Furthermore, $C_2H_2$-TPD was employed to evaluate the $C_2H_2$ gas enrichment ability of Cu-PCC[42]. The greater $C_2H_2$ adsorption on Cu-PCC than on its Cu-C counterpart, along with the ever-increasing desorption temperature under similar specific surface areas (Fig. 3h and Supplementary Fig. 17), indicated that the

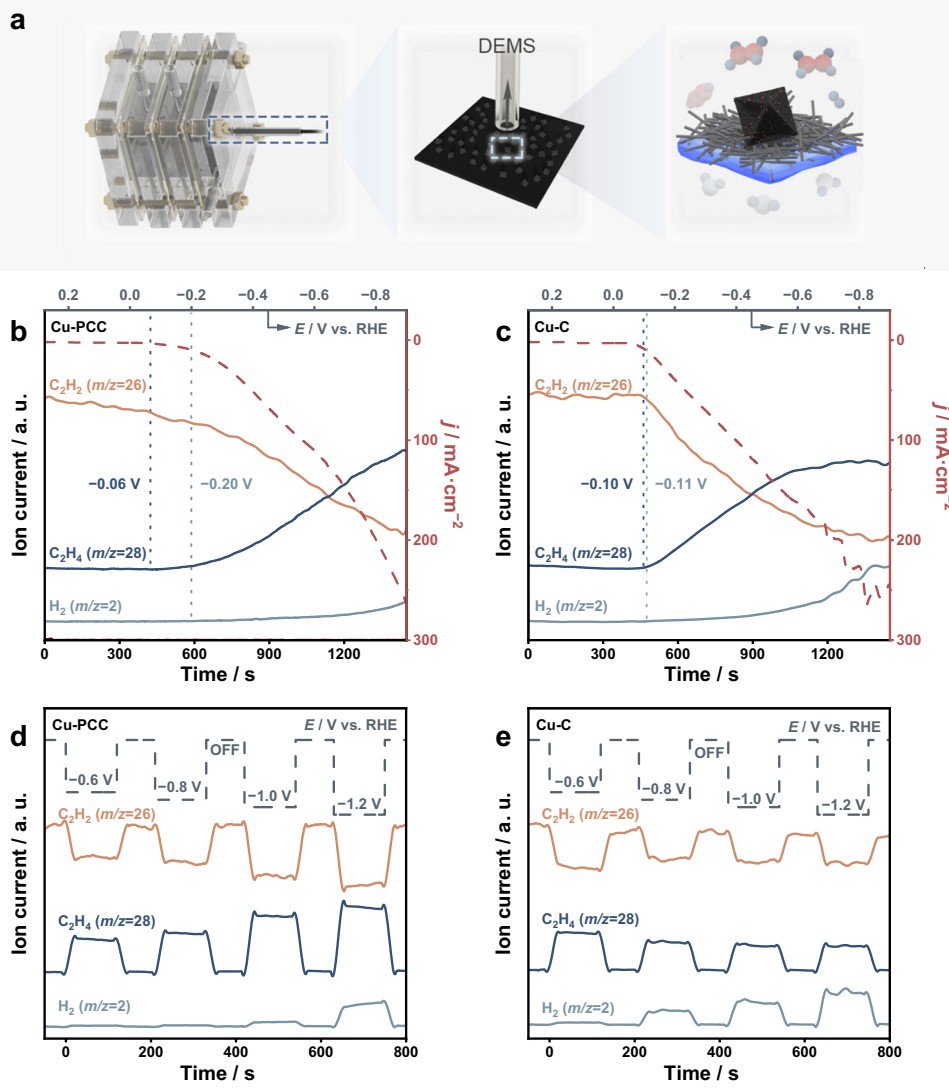

**Fig. 4 | ESAE reaction analysis over Cu-PCC and Cu-C. a** Schematic illustration of the equipment used in the DEMS analysis. **b**, **c** LSV curves and the corresponding DEMS signals of Cu-PCC and Cu-C. **d**, **e** Square wave potentials and the corresponding DEMS signals of Cu-PCC and Cu-C. *R*: $1.5 \pm 0.1 \, \Omega$.

enriched $C_2H_2$ and optimized mass transfer were the main reasons for the enhanced interactions between the $C_2H_2$ feedstocks and Cu-PCC.

## ESAE reaction analysis and performance evaluation

The ESAE process was evaluated in a three-electrode flow cell with a gas diffusion layer under potentiostatic conditions using simulated raw coal-derived $C_2H_2$ (-15%) as the feeding gas (Supplementary Fig. 18a). First, online differential electrochemical mass spectrometry (DEMS) was conducted under linear sweep voltammetry (LSV) mode to explore and analyse the ESAE process (Fig. 4a). In addition to Cu-PCC possessing a more positive onset potential for $C_2H_2$ hydrogenation (−0.06 V vs. RHE) than Cu-C (−0.1 V vs. RHE), Cu-PCC has a much more negative HER onset potential, endowing Cu-PCC with a better ability to activate $C_2H_2$ and suppress the HER (Fig. 4b, c and Supplementary Figs. 18–20). Moreover, the signal intensity of $C_2H_4$ over Cu-C displays a nearly volcanic shape, and the response of $H_2$ acutely increases under potentials more negative than −0.6 V vs. RHE. However, the $C_2H_4$ signal always dominated the whole product until the end of the LSV over Cu-PCC. In other words, the production gap between $C_2H_4$ and the $H_2$ byproduct becomes larger with decreasing potential over Cu-PCC, while it shows an inverse trend over Cu-C, further verifying the superiority of Cu-PCC in the ESAE process. In addition, the same

electron transfer number of $C_2H_2$ to $C_2H_4$ and $H_2O$ to $H_2$ is likely the reason for the similar LSV curves of the two catalysts. Then, we performed the DEMS test under square wave potentials. For Cu-PCC, $H_2$ emerges under a much more negative potential than does its Cu-C counterpart. In addition, unlike the almost unchanged or even decreased $C_2H_4$ signal observed for Cu-C, the $C_2H_4$ signal increases with decreasing potential, indicating better $C_2H_4$ selectivity in Cu-PCC under a raw coal-derived $C_2H_2$ atmosphere (Fig. 4d, e).

The quantification of the ESAE process showed that the FE of $C_2H_4$ over Cu-PCC exceeded -90%, and the $C_2H_6$ byproduct was almost undetectable throughout the whole range (Fig. 5a and Supplementary Fig. 21). However, the FE of $C_2H_4$ decreased rapidly, with more $H_2$ produced at more negative potentials than −0.8 V vs. RHE over Cu-C (Fig. 5b). Furthermore, the obtained $C_2H_4$ production rate of Cu-PCC at a potential of −1.2 V vs. RHE was 3.42 mol $g_{cat}^{-1}$ $h^{-1}$ with a partial current density and $C_2H_2$ conversion of 0.42 A cm$^{-2}$ and 56.31%, respectively, greatly surpassing those of its Cu-C counterpart (2.00 mol $g_{cat}^{-1}$ $h^{-1}$, 0.26 A cm$^{-2}$ and 33.21% $C_2H_2$ conversion) and the recently reported catalysts (Fig. 5a, b and Supplementary Table 1). The superiority of the $C_2H_4$ production rates of Cu-PCC became more obvious after normalization by the electrochemical surface area (ECSA) or Cu loading capacity (Fig. 5c and Supplementary Figs. 22–24). In addition, the

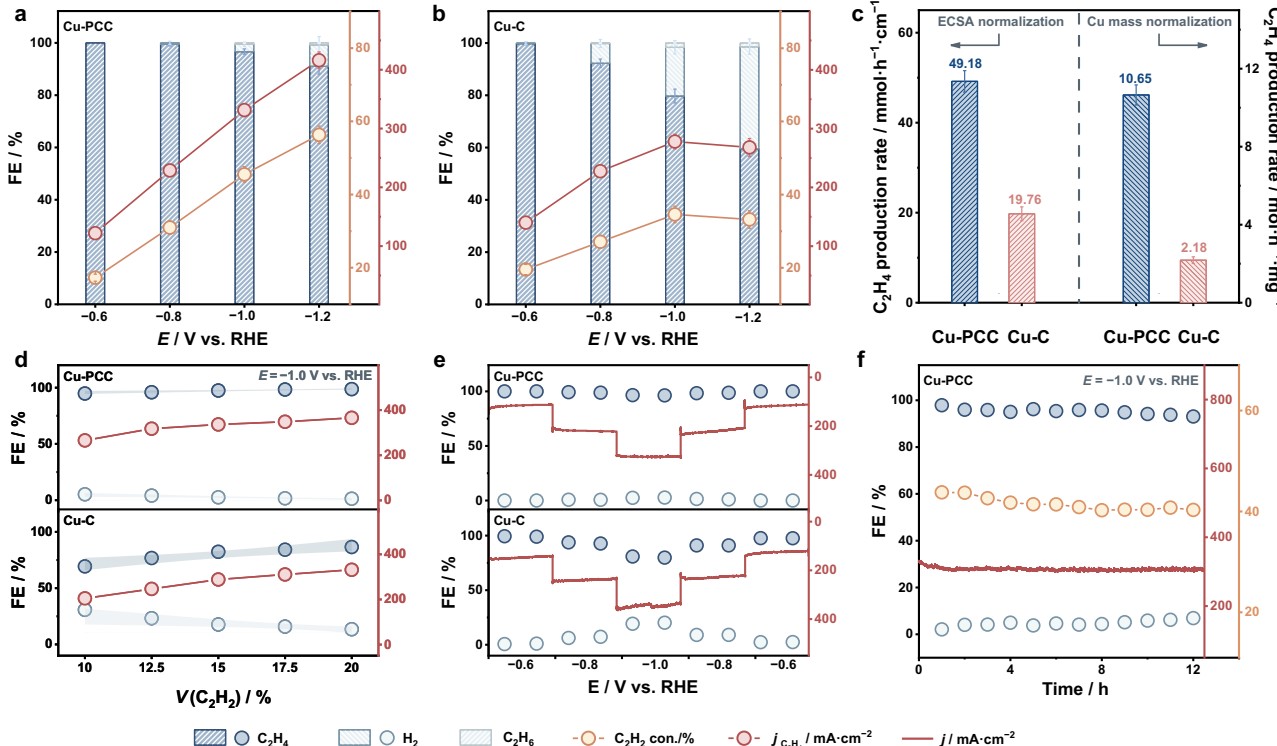

**Fig. 5 | Performance evolution. a, b** Potential-dependent conversion (con.) of $C_2H_2$ and FE of the obtained products over Cu-PCC and Cu-C. **c** The $C_2H_4$ yield normalized to the ECSA and Cu loading capacity. **d** Performance evaluation under different $C_2H_2$ concentrations. (The shaded areas represent the 95% confidence intervals.) **e** Potential fluctuation test of Cu-PCC and Cu-C. **f** Continuous test of the Cu-PCC. The error bars correspond to the standard deviation of at least three independent measurements, and the center value for the error bars is the average of the three independent measurements.

better performance of Cu-PCC over its slightly larger particle size further demonstrates the superiority of the nanosized concave surface (Supplementary Fig. 10). In addition, considering the demand for industrial production, stability evaluation experiments at various concentrations and step potentials for the hydrogenation of $C_2H_2$ were performed. Accordingly, the 95% confidence intervals of the FEs were calculated to evaluate the selectivity stability, as shown in Fig. 5d. The narrower confidence intervals of the $C_2H_4$ and $H_2$ FEs over Cu-PCC than Cu-C demonstrate that the fitted lines of FEs under different concentrations of Cu-PCC are more precise[43], which means that the influence of concentration on FEs is less significant over Cu-PCC[44], indicating its better stability. In addition, the FE and selectivity of $C_2H_4$ remain unchanged at different potentials over Cu-PCC, which is superior to its counterpart (Fig. 5e). These results indicate that the proposed Cu-PCC exhibits potential- and concentration-independent ESAE activity, which is suitable for practical application. Note that both the FE of $C_2H_4$ and the $C_2H_2$ conversion over Cu-PCC remained unchanged within the error range during the 12 h continuous test, suggesting its robust durability (Fig. 5f and Supplementary Fig. 25).

**Mechanistic exploration of the high selectivity for ethylene**
To elucidate the reason for the increase in $C_2H_4$ FE and selectivity over Cu-PCC under a raw coal-derived $C_2H_2$ atmosphere, a series of characterizations were applied. First, kinetic isotope effect (KIE) measurements show that the $k_H/k_D$ values among the selected potentials over Cu-PCC and Cu-C are all in the range of 1 - 2 with a similar trend (Supplementary Fig. 26), indicating that the hydrogen-reliance nature of the ESAE process did not change due to the concave support. Then, in situ attenuated total reflectance−Fourier transform infrared (ATR − FTIR) and Raman spectroscopy were used to evaluate the status and coverage of $C_2H_2$ with the catalytic surface (Supplementary Figs. 27–29). Generally, the peak frequency of the IR or Raman

characteristic peak is determined by the strength of the corresponding bond[45–48]. For $C_2H_2$, the enhanced $\pi$ conjugation led to the redistribution of bonding electrons, and the corresponding $\pi$ bond was weakened due to the delocalization of electrons, consequently leading to a negative shift in the peak frequency (redshift)[49]. As shown in Fig. 6a, b, both $\nu(C − H)$ ( ~ 3200 cm$^{-1}$) and $\nu(C ≡ C)$ ( ~ 1620 cm$^{-1}$) of $C_2H_2$ over Cu-PCC shift to lower frequencies than do their Cu-C counterparts at each potential (Supplementary Figs. 30–31)[7,8,50,51], indicating that the triple bond of $C_2H_2$ becomes unstable over Cu-PCC due to the delocalization of $\pi$ electrons; thus, the $C_2H_2$ molecule is easier to activate. Moreover, the redshift of the peak attributed to adsorbed $C_2H_2$ in the Raman spectrum from 1700 cm$^{-1}$ over Cu-C to 1685 cm$^{-1}$ over Cu-PCC also confirmed the attenuation of the $\pi$ bonds of $C_2H_2$, further demonstrating that support with concave surfaces is beneficial for $C_2H_2$ activation (Fig. 6c, d and Supplementary Fig. 32)[20,21]. In addition, considering that the integrals of the IR bands are related to the coverage of the respective adsorbate on the surface, the area ratio between $\nu(C ≡ C)$ and $\delta(H-O-H)$ was viewed as the descriptor of the relative coverage of $C_2H_2$ over the catalyst surface. The plot of $\nu(C ≡ C)/\delta(H − O − H)$ over Cu-C exhibited a volcano-like profile, which began to decrease at potentials more negative than −0.2 V vs. RHE, indicating that $H_2O$ adsorption improved with the negative shift potential and accounted for the strong HER competition. Conversely, the plot of $\nu(C ≡ C)/\delta(H − O − H)$ over Cu-PCC presented a nearly monotonically increasing trend with negatively shifted potentials (Fig. 6e), demonstrating the higher $C_2H_2$ coverage of Cu-PCC under the applied potential range and accounting for the enhanced intermolecular $\pi$ conjugation and the delocalization of $\pi$ electrons from $C_2H_2$[52]. Finally, we also conducted DFT calculations to evaluate the $C_2H_2$ hydrogenation energy barrier under high and low coverage to verify our experimental results. As shown in Fig. 6f, the hydrogenation barriers under high $C_2H_2$ coverage are lower than those under low coverage

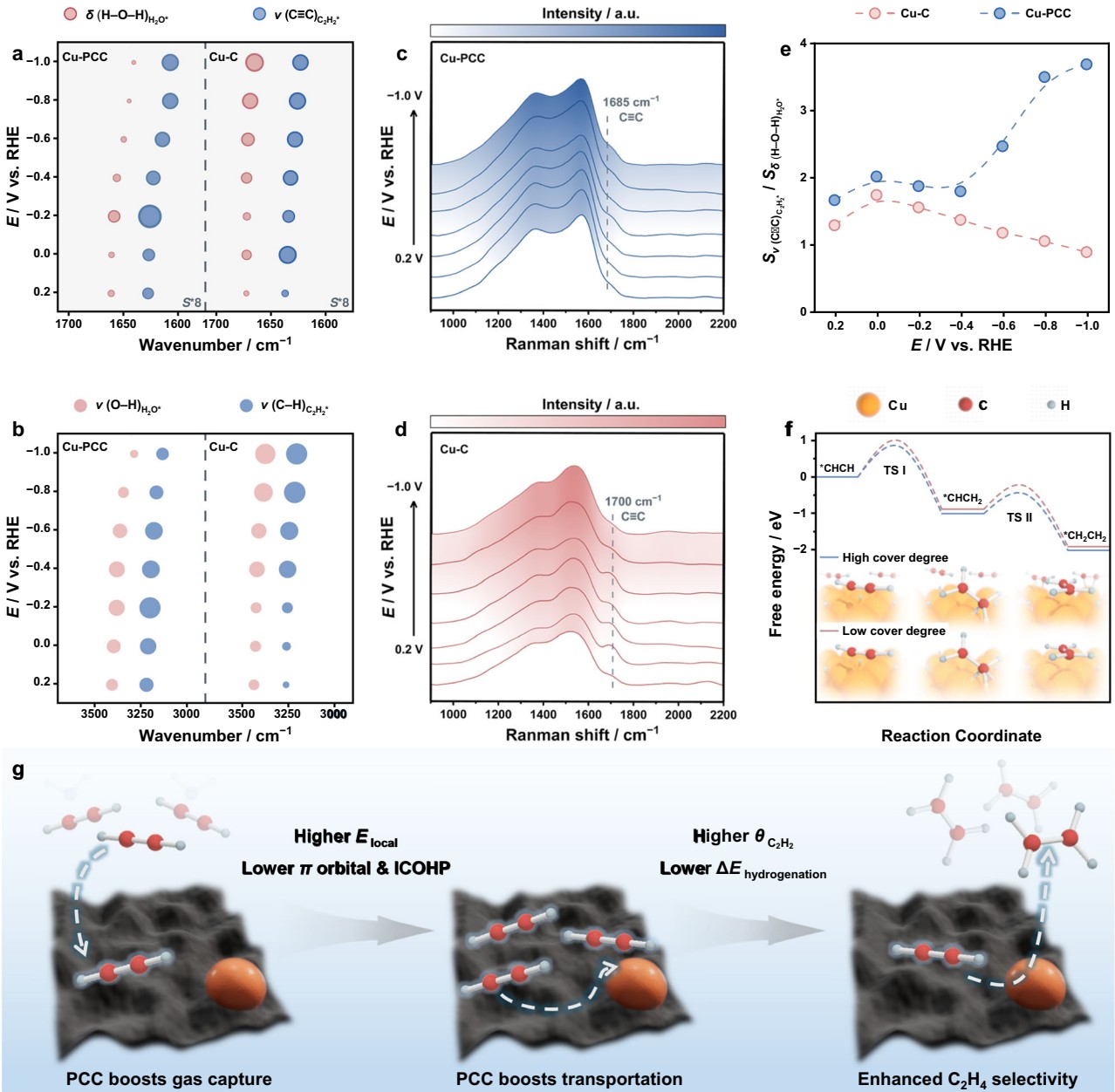

**Fig. 6 | Exploration of the origin of performance enhancement. a, b** Position and intensity comparison of the characteristic peaks of potential-dependent in situ ATR-FTIR spectra of $H_2O$ and $C_2H_2$ species over Cu-C and Cu-PCC. **c, d** In situ Raman spectra of Cu-PCC and Cu-C using $C_2H_2$ as the feeding gas. **e** The area ratio of adsorbed $C_2H_2$ to $H_2O$ at different potentials over Cu-PCC and Cu-C. **f** Free-energy diagram for the $C_2H_2$ hydrogenation process under different $C_2H_2$ coverages. **g** Schematic illustration of the mechanism for the enhanced $C_2H_4$ selectivity over Cu-PCC.

(Supplementary Figs. 33−37), indicating easier activation of $C_2H_2$ and better hydrogenation kinetics over Cu-PCC. Accordingly, the in-depth origins for the enhanced $C_2H_4$ selectivity obtained using low-concentration raw coal-derived $C_2H_2$ over Cu-PCC are summarized in Fig. 6g. The $C_2H_2$ feedstocks could be enriched by the nanosized concave carbon surfaces and then effectively transferred to the Cu sites, consequently resulting in high $C_2H_2$ partial pressure and coverage. Then, the electron delocalization effect due to the increased $C_2H_2$ coverage promoted $C_2H_2$ activation, thus leading to satisfactory $C_2H_4$ selectivity and FE over Cu-PCC.

## Discussion

In summary, Cu nanoparticles loaded on carbon supports with abundant nanosized concave surfaces were designed to enhance $C_2H_2$ adsorption for direct utilization of raw coal-derived $C_2H_2$. Cu-PCC delivered a $C_2H_4$ FE of 91.7% and a single-pass $C_2H_2$ conversion of 56.31% under a potential of −1.2 V vs. RHE at a partial current density of 0.42 A cm⁻², greatly outperforming its counterpart without the concave surface. Notably, the nanosized concave surfaces were significantly enriched in $C_2H_2$ gas and had lower mass transfer kinetics, resulting in higher $C_2H_2$ coverage. Moreover, the delocalization of π electrons in $C_2H_2$ due to the strengthened intermolecular π conjugation caused by the increased $C_2H_2$ coverage promoted the activation of $C_2H_2$, thus endowing Cu-PCC with robust HER suppression ability and better $C_2H_4$ selectivity. Our work may not only demonstrate an efficient and selective catalyst for nonpetroleum $C_2H_4$ electrosynthesis but also open a facile way to access low-concentration gaseous reactants for various catalytic applications.

## Methods

### Materials

All chemicals used in the experiments were analytically pure and used without further purification. Copper nitrate ($Cu(NO_3)_2 \cdot 3H_2O$, 99.0%), N,N-dimethylformamide (DMF), polyvinyl pyrrolidone (PVP, $M_w = 15,000–19,000$), tannic acid (TA, 99.0%), anhydrous ethanol ($CH_3CH_2OH$, 99.9%), and 1,3,5-benzene tricarboxylic acid ($H_3BTC$) were purchased from Aladdin Ltd. (Shanghai, China). Simulated raw coal-derived $C_2H_2$ with concentrations of ~15% and 3% $H_2$/Ar were purchased from Lian Bo (Tianjin, China) Co., Ltd. A Nafion 117 exchange membrane with a thickness of 183 μm was purchased from DuPont and used after sequent heat treatments in 5% $H_2O_2$ and $H_2SO_4$ solutions under 80 °C. The Hg/HgO reference electrode (diameter 1.8 mm) and the gas diffusion electrode (GDE) (Carbon paper 29BC) was purchased from Shanghai Chuxi Industrial Co., Ltd. Deionized water (DIW) was used in all the experimental processes.

### Synthesis of Cu-MOF precursors

According to previous literature[37], the Cu-MOF precursors were prepared by a PVP-assisted strategy as follows. First, 1.46 g of $Cu(NO_3)_2 \cdot 3H_2O$ and 0.7 g of $H_3BTC$ were dissolved in 20 mL of DMF to form solution A and solution B, respectively. Subsequently, 0.5 g PVP was added to solution A and stirred for 5 min to obtain a homogenous solution. Then, solution B was mixed with solution A and stirred for an additional 10 min. Afterward, the mixture was transferred to a 100 mL Teflon-lined stainless-steel autoclave and maintained at 80 °C for 24 h. Finally, the blue precipitates were harvested by centrifugation, washed with DIW and ethanol several times, and dried in a vacuum oven overnight to produce the Cu-MOF precursors.

### Synthesis of Cu-TA

The as-prepared Cu-MOF precursors (100 mg) and tannic acid (TA) (50 mg) were first dispersed into 50 mL of DIW to form two solutions. The two solutions were subsequently mixed at room temperature and stirred for 30 min. Afterward, the mixture was put into an oil bath at 50 °C and refluxed under continuous magnetic stirring (stirring speed: 700 rpm) for 7 h. The precipitate was then washed with DIW and absolute ethanol at least three times to remove the residual TA and dried at 70 °C in a vacuum oven overnight.

### Synthesis of Cu-PCC and Cu-C

To obtain Cu-PCC and Cu-C, the as-prepared Cu-TA and Cu-MOF precursors were annealed at 400 °C for 2 h at a heating rate of 5 °C min$^{-1}$ under a 3% $H_2$/Ar atmosphere. The mixture was then naturally cooled to room temperature.

**Fabrication of Cu-PCC and Cu-C electrodes.** The electrodes used in this work were fabricated by the traditional spin-casting method. The commercial GDE was cut into a square shape with a size of $1.2 \times 1.2$ cm$^2$ as the electrode substrate. Specifically, Cu-PCC and Cu-C were dissolved in a mixed solvent of water and ethanol with a volume ratio of 1/3 to form a solution at a concentration of 2 mg mL$^{-1}$, respectively. Then, the as-prepared solutions were spin-coated on the GDE substrates with 1 mL on each substrate under a constant spin speed of 500 rpm to obtain the electrodes with a loading of 1 mg Cu-PCC and/or Cu-C.

### General characterizations

Quasi-in situ powder X-ray diffraction (XRD) was performed on a Bruker D8 Focus Diffraction System (Germany) using a Cu $K\alpha$ radiation source ($\lambda = 0.154178$ nm). Scanning electron microscopy (SEM) and scanning transmission electron microscopy (STEM) were conducted with an FEI Apreo S LoVac microscope (10 kV). Transmission electron microscopy (TEM) and high-resolution transmission electron microscopy (HRTEM) images were obtained with a

JEOL-2100F system equipped with an EDAX Genesis XM2. X-ray photoelectron spectroscopy (XPS) was conducted with a PHI-1600 X-ray photoelectron spectrometer equipped with Al $K\alpha$ radiation. All the peaks were calibrated with the Ti 2p spectrum since C 1 s is a key parameter in our research. The Raman spectra were obtained with a Renishaw inVia reflex Raman microscope under excitation with a 514 nm laser at a power of 20 mW. Fourier transform infrared spectroscopy (FTIR) was performed on a Nicolet IS50 instrument. The Brunauer–Emmett–Teller (BET) surface area was measured by $N_2$ adsorption using a Micromeritics ASAP 2460. Inductively coupled plasma–optical emission spectrometry (ICP–OES) was conducted with an Agilent 5110 instrument (OES). Atomic force microscopy (AFM) was carried out on a Bruker Dimension Icon.

### Electrochemical measurements in the flow cell

Electrochemical measurements were carried out in a typical flow cell consisting of a GDE as the working electrode, Pt foil as the counter electrode, and Hg/HgO (Note that the Hg/HgO electrode was calibrated with respect to a reversible hydrogen electrode in a high-purity hydrogen-saturated electrolyte with a Pt foil as the working electrode.) as the reference electrode using a CS150H electrochemical workstation. The volume of each compartment is around 1.5 mL. In addition, the potentials were scaling to RHE using Eq. (1). The cathode cell and anode cell were separated by a Nafion 117 proton exchange membrane. The cathode and anode electrolytes were both composed of 1.0 M freshly prepared KOH solution, of which the pH value is around $13.6 \pm 0.3$, and a peristaltic pump was used to circulate the liquid phase. The gas flow rate was controlled by a mass flowmeter. Before the performance tests, the working electrode was fixed at the interface between the gas flow block and the cathodic electrolyte block by conductive copper tape. First, the electrochemical semihydrogenation of acetylene was conducted at different applied potentials for 10–20 min to achieve relatively stable and reliable performance parameters before quantitative analysis. The gas at the flow cell outlet was directly introduced into the gas chromatography system for analysis of the products. Before LSV, the resistance ($R$) was measured firstly using CS150H electrochemical workstation and the $R$ values were $1.5 \pm 0.1$ Ω. Only the LSV curves provided in this work were $iR$ compensated with a compensation level of 70%. For the Tafel slopes, the LSV curves were replotted by using the logarithms of the current density as the x-axis and the potential as the y-axis. The obtained slopes of the linear part of the replotted figure were the Tafel slopes. After LSV process, the electrolysis was conducted under a potentiostatic mode in the range from −0.6 to −1.2 V vs. RHE.

$$E \text{ vs. RHE} = E \text{ vs. Hg/HgO} + E^{\theta}(\text{Hg/HgO}) + 0.0591 \times \text{pH} \qquad (1)$$

### Quantitative analysis of the $C_2H_2$ conversion, evolution rate, and FE of the obtained products

The products were subjected to a GC – 2010 gas chromatograph equipped with an activated carbon-packed column (with He as the carrier gas) and a barrier discharge ionization detector. The $C_2H_2$ conversion and evolution rate of the different products were calculated using Eqs. (2)–(5), and the FEs of the different products were calculated using Eq. (6). All the experiments were repeated three times.

$$C(\text{X}) = k(\text{X}) \times \text{peak area} \qquad (2)$$

$$n(\text{X}) = C(\text{X}) \times V \qquad (3)$$

$$\text{Conversion (\%)} = \frac{n(C_2H_2)_{\text{in}} - n(C_2H_2)_{\text{out}}}{n(C_2H_2)_{\text{in}}} \times 100\% \qquad (4)$$

$$\text{Evolution Rate (mmol/mg/h)} = \frac{n(X)}{m} \times S \qquad (5)$$

$$FE_X(\%) = \frac{a \times n(X) \times F}{Q} \qquad (6)$$

X: The feedstock and products, including $C_2H_2$, $H_2$, $C_2H_4$, and $C_2H_6$.

C: The concentration of feedstock and products.

m: The mass of catalysts over the electrode.

n: The moles of feedstock and products.

k: The slope of the calibration curves for feedstock and products.

S: The gas flow rate.

a: The electron transfer number.

F: Faraday constant.

Q: The total Coulomb number of the ESAE process.

### Electrochemical operando online DEMS analysis

Operando online DEMS analysis was conducted with a QAS 100 instrument provided by Linglu Instruments (Shanghai) Co., Ltd. Because the products in the proposed ESAE process were all in the gas phase, *operando* experiments were conducted to monitor the distribution of the products during the on-stream reaction, clarifying the selectivity issues more directly and clearly. The DEMS was conducted in the same flow cell electrolyzer with our performance evaluation to ensure that the gas at the flow cell outlet was directly injected into the negatively pressured gas circuit system of the DEMS through a quartz capillary that was inserted into the outlet of the flow cell and the schematic of DEMS testing has been provided in Supplementary Fig. 18. In addition, the membrane employed was only for the separation of cathode and anode using Nafion 117 proton exchange membrane. Note that all the ion currents plotted in this work are provided without any correction or subtraction. The LSV test and rectangular wave potentials were applied from 0.3 to −1.2 V vs. RHE with a constant interval of 400 s using a CS150H electrochemical workstation. During the experiment, the flow rates of $C_2H_2$ gas and the electrolyte were set the same as those used for the performance evaluation.

### Electrochemical in situ ATR-FTIR measurements

In situ ATR-FTIR was performed on a Nicolet 6700 FTIR spectrometer equipped with an MCTA detector with silicon as the prismatic window and an ECIR-II cell by Linglu Instruments. First, Cu-PCC was carefully dropped on the surface of the gold film, which was chemically deposited on the surface of the silicon prismatic material before each experiment. Then, the deposited silicon prismatic material served as the working electrode. Pt foil and Hg/HgO with an internal reference electrolyte of 1.0 M KOH were used as the counter and reference electrodes, respectively. A 1 M KOH solution was used as the electrolyte. The electrolyte was presaturated with pure $C_2H_2$ gas, and the gas was continuously bubbled through during the whole measurement. The spectrum was recorded every 30 s under an applied potential ranging from 0.2 to −1.0 V vs. RHE.

### Electrochemical in situ Raman measurements

In situ electrochemical Raman spectra were recorded via an electrochemical workstation on a Renishaw inVia reflex Raman microscope under 532 nm laser excitation under controlled potentials. We used a homemade Teflon electrolytic cell equipped with a piece of round quartz glass for the incidence of lasers and protection of the tested samples1. Before the experiments, the electrolyte was pretreated with pure $C_2H_2$ gas to obtain $C_2H_2$-saturated KOH. The working electrode was parallel to the quartz glass to maintain the plane of the sample

perpendicular to the incident laser. The Pt wire was rolled to a circle around the working electrode to serve as the counter electrode. The reference electrode was Hg/HgO with an internal reference electrolyte of 1.0 M KOH. The spectrum was recorded under applied potentials ranging from 0.2 to −1.0 V vs. RHE.

### Computational details

All the DFT calculations were performed using the Vienna ab initio simulation package (VASP) (Supplementary Data 1 for the optimized DFT computational models)[53]. The projector augmented wave (PAW) pseudopotential with the PBE generalized gradient approximation (GGA) exchange-correlation function was utilized in the computations[54,55]. The cut-off energy of the plane wave basis set was 500 eV, and a Monkhorst-Pack mesh of 3×3×1 was used in K-sampling for the adsorption energy calculations and other nonself-consistent calculations. The long-range dispersion interaction was described by the DFT-D3 method. The electrolyte was incorporated implicitly with the Poisson-Boltzmann model implemented in VASPsol[56]. The relative permittivity of the media was chosen to be $\epsilon_r = 78.4$, corresponding to that of water. All atoms were fully relaxed with an energy convergence tolerance of $10^{-5}$ eV per atom, and the final force on each atom was <0.05 eV Å$^{-1}$.

The transition state (TS) searches were performed using the Dimer method in the VTST package. The final force on each atom was <0.1 eV Å$^{-1}$. The TS search is conducted by using the climbing-image nudged elastic band (CI-NEB) method to generate initial guess geometries, followed by the dimer method to converge to the saddle points.

In the ab initio molecular dynamics (AIMD) simulations, canonical ensemble (NVT) conditions were imposed by a Nose–Hoover thermostat with a target temperature of 300 K. The MD time step was 1 fs, and all the systems were run for 10 ps to reach equilibrium. In the plane and angle models, 114 and 120 water molecules were added to ensure that the density of water in the model was approximately 1 g/cm³. The last 1 ps of data in the AIMD process are selected for analysis. In the process of hydrogen bond analysis, we set the maximum distance of the hydrogen bond to 3.5 Å and the angle cut-off to 40°.

The adsorption energy of the reaction intermediates can be computed using Eqs. (7)−(8):

$$\Delta E = E_{*ads} - (E_* + E_{ads}) \qquad (7)$$

$$\Delta G = \Delta E + \Delta E_{ZPE} - T\Delta S \qquad (8)$$

where $\Delta E_{ZPE}$ is the zero-point energy change and $\Delta S$ is the entropy change. In this work, the values of $\Delta E_{ZPE}$ and $\Delta S$ were obtained via vibration frequency calculations.

## Data availability

The data that support other plots within this paper are available from the corresponding author upon request. The source data underlying Figs 1–6 are provided as a Source Data file. Source data and the optimized DFT computational models are provided with this paper. Source data are provided with this paper.

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

## Acknowledgements

We acknowledge the National Natural Science Foundation of China (22271213 to B.Z. and 22209120 to B.-H.Z.) and the Fundamental Research Funds for the Central Universities of China.

## Author contributions

B.Z. conceived the idea and directed the project. F.C., B.-H.Z., and B.Z. designed the experiments. L.L. and F.C. carried out the experiments and characterization. Y.Y. assisted in some experiments. C.C. performed the DFT calculations. F.C. and B.-H.Z. wrote the paper. B.Z. revised the paper. All the authors discussed the results and commented on the paper.

## Competing interests

The authors declare no competing interests.
