## [Peer Review File · Nature Communications]

REVIEWER COMMENTS

Reviewer #1 (Remarks to the Author):

In this work, the authors developed a porous concave carbon-supported Cu nanoparticle (Cu-PCC) electrode for ethylene electrocatalysis through the direct hydrogenation of the low-concentrated raw coal-derived acetylene. Here, they demonstrated that the enriched concentration of C_2H_2 and boosting mass transfer kinetics are critical for the suppressed hydrogen evolution reaction competition and thus leading to the enhanced selectivity of the target C_2H_4 even if under the relative low concentration raw-derived coal-derived C_2H_2 . Moreover, a series of in situ characterizations further provided insight into the advantages in C_2H_2 activation of their proposed Cu-PCC. In general, this work is well done and a solid contribution to the emerging area of the acetylene industry and I support its eventual publication in Nature Communications, given the authors address several concerns below.

1. Firstly, since Cu-C and Cu-PCC were transformed from different Cu-based complexes, will the surface oxygen-containing groups extents of these two carbon supports vary? And if these differences could contribute to the enhanced performance? The author fails to provide such an important characterization.

2. Similar to the concerns raised above, I wonder if the hydrophilicity of Cu-PCC is varied from the Cu-C counterpart due to its unique nanosized concave surfaces. If so, will the modulated hydrophilicity account for the enriched C_2H_2 ?

3. For performance evaluation, the reviewer noticed that there is a small fraction of C_2H_6 exists both in Cu-PCC and Cu-C, however, the signal of C_2H_6 in the online differential electrochemical mass spectrometry is absent without explanation.

4. The following is not criticism, just discussion:

Considering that the integrals of the IR bands could be related to the coverage of the respective adsorbate on the surface, using the area ratio to describe the relative coverage of two species is quite an insightful method, however, the recognition of specific species should be cautious. I have one technical question on how the author distinguishes the characteristic peaks of H_2O and C_2H_2 ? I think it deserves to be written in the Supporting Information for the judgment of such a method.

Reviewer #2 (Remarks to the Author):

This manuscript proposes a design strategy using nanosized concave to regulate the interfacial microenvironment, which increase the local C_2H_2 concentration and optimize its mass transfer kinetics. Thus, the electrocatalytic semihydrogenation of raw coal-derived C_2H_2 performance over the as-designed Cu-PCC is greatly enhanced and enables a high C_2H_4 Faradaic efficiency and C_2H_2 single-pass conversion. In situ ATR-SEIRAS demonstrates the C_2H_2 molecule is enriched around the Cu-PCC interface and becomes easier to be activated, resulting in outstanding performance. In summary, this work is very interesting and coherent. Therefore, I recommend the acceptance of this manuscript after clarifying my

concerns that listed below.

1. The author claimed that the adsorption of C₂H₂ on Cu is greater than that on carbon supports without specific comparison. That is there is no specific value of ICOHP of the acetylene molecule adsorbed over Cu in supplementary figure 1.
2. The error bar in Fig. 5c is missing. It is suggested that the author supplement the experiment and improve credibility of Fig. 5c.
3. The particle size of catalytic materials can significantly impact performance, and even induce different reaction mechanisms. The authors should provide statistical distributions of particle sizes for the two materials.
4. In Figure 6c and d, the authors demonstrate slight differences in the Raman shift of acetylene on Cu-PCC and Cu-C. However, it should be careful when analyzing Raman signals in the range of 1000-1800 cm⁻¹ due to that the skeletal signals of carbon materials also appear at this range. The authors should further clarify how they identify signals near 1700 cm⁻¹ and ensure that they originate from the reactant acetylene rather than the reduced carbon carrier.
5. The description of IR and Raman characterization is too vague to be compared with other work. More details regarding signal acquisition should be provided for its reference value.

Reviewer #3 (Remarks to the Author):

Seeking a green and petroleum-independent alternative toward ethylene synthesis is significant to the manufacturing industry of commodity chemicals, medicines, and agrochemicals. This manuscript demonstrated the electrocatalytic hydrogenation of acetylene derived from coal produce ethylene and focused on directly using the low concentration raw coal-derived C₂H₂ as feedstocks for further decrease the costs of C₂H₄ production. They proposed and confirmed that the negative polarization field caused by the concave surface enhances the conjugate action between C₂H₂ and the carbon support, which is suitable for C₂H₂ enrichment. Besides, the unique concave support could also boost the mass transfer kinetics of C₂H₂ toward Cu sites. This interesting work is an impressive breakthrough in the petroleum-independent C₂H₄ electrosynthesis, thus I would like to recommend this interesting work for being accepted after addressing the following minor issues:

- (1) The author proposed a sequential self-template transformation method based on the Kirkendall effect to construct Cu-PCC. However, the details of TA exchange process is missing.
- (2) As for the X-ray absorption spectroscopy results in Supplementary Figures 11b, c, it is meaningless to provide only the comparison of Cu-PCC and Cu-C without the standard results of Cu foil.
- (3) The whole manuscript focused on the modulation of C₂H₂ feedstock, however, H₂O, as a hydrogen source, should also be considered. As the recently reported electrocatalytic hydrogenation cases, the kinetic isotope effect experiment should be carried out to

supplement the reaction mechanism.

(4) Additionally, I have noticed that the authors claim that their products were quantified using GC-2010 gas chromatography, but I did not find any information of the method they applied. Is that the calibration curve method? Or internal standard method? The author should pay much attention to such details.

(5) There are also some format and images display issues in the present manuscript which should be addressed, such as Figures 4b, c, and d (there are strange lines exist), Supplementary Figure 8c.

(6) The performance of C₂H₄ production from electrocatalytic semihydrogenation of C₂H₂ of this work should be compared with the recent reported catalysts to present its significance and innovation.

(7) The characterization of the Cu-PCC after electrocatalytic stability should be provided to support its long-term stability and confirm its true active sites.

Reviewer #4 (Remarks to the Author):

In this manuscript, the authors focus on the raw coal-derived C₂H₂ semi- hydrogenation process and design a porous concave carbon-supported Cu nanoparticle (Cu-PCC) restraining the competing HER. The well-designed carbon carrier with nanosized concave is synthesized through a facile self-template and benefits for the mass transport and the activation of C₂H₂. Using a low concentration of C₂H₂, the Cu-PCC deliver a high C₂H₄ Faradaic efficiency and a good single-pass C₂H₂ conversion at -1.2 V vs RHE, outperforming the counterpart without concave surface. The origin of performance enhancement is also clarified. I believe that this work is highly recommended for its publication in Nature Communications after minor modifications.

(1) Parameters used in the molecular dynamics simulation calculations should be explicitly stated as they can significantly influence the simulation results.

(2) The comparison of the distance between water clusters and the catalysts layer in Figure 2d is intuitive, yet the authors have not conducted further quantitative analysis. Supplementing this discussion would make the conclusion more comprehensive.

(3) The author did not include cross-section data in Figure 3f. It is suggested that the author supplement these data and the raw format of AFM data.

(4) Cu Auger LMM spectra, which could provide more sensitive surface valence state information compared to Cu 2p spectra, should be added to eliminate the activity contribution from electronic structures differences.

(5) There are some minor formatting and spelling issues in the text, such as in Fig. S25a, where the x-axis of "energy change profiles" should be "reaction coordinate"; the significant figure precision of ICOHP in Fig. 2b. Additionally, there are instances of minus and hyphen misuses. The author should carefully review the manuscript.

A point-by-point response to the reviewers' comments

To reviewer 1:

Reviewer letter: In the submitted manuscript, the authors developed a porous concave carbon-supported Cu nanoparticle (Cu-PCC) electrode for ethylene electrosynthesis through the directly hydrogenation of the low-concentrated raw coal-derived acetylene. Here, they demonstrated that the enriching concentration of C₂H₂ and boosting mass transfer kinetics are critical for the suppressed hydrogen evolution reaction competition and thus leading to the enhanced selectivity of the target C₂H₄ even if under the relative low concentration raw-derived coal-derived C₂H₂. Moreover, a series of in situ characterizations further provided insight into the advantages in C₂H₂ activation of their proposed Cu-PCC. In general, this work is well done and a solid contribution to the emerging area of acetylene industry and I support its eventual publication in Nature Communications, given the authors address several concerns below.

Answer: Thank you very much for your constructive suggestions to improve the quality of our work and for your positive comments on our revised manuscript. Regarding the concerns of the reviewer, we have provided a point-by-point response. To save the reviewer's valuable time, key revisions are displayed on a yellow background in the revised manuscript and Supplementary Information (SI). We are sure that the quality of this work will be greatly improved after being revised.

Comment 1. Firstly, since that Cu-C and Cu-PCC were transformed from different Cu-based complexes, will the surface oxygen-containing groups extents of these two carbon supports varies? And if these different could contribute to the enhanced performance? The author fails to provide such an important characterization.

Answer 1: We acknowledge the reviewer's constructive comments and suggestions. We apologize for the carelessness in that we only provided the Raman results of Cu-PCC in our original version. According to the reviewer's suggestion, the IR spectrum, which is sensitive to oxygen-containing groups, was obtained for comparison of Cu-PCC and Cu-C in the revised Supplementary Information. The results show that there is no difference in the extent of oxygen-containing groups between Cu-C and Cu-PCC (Figure R1 or Supplementary Figure 14b in the revised Supplementary Information). To save the reviewer's valuable time, key revisions are displayed on a yellow background in the revised Supplementary Information.

Figure R1 IR spectra of Cu-PCC and Cu-C at OCP.

Comment 2. Similar to the concerns raised above, I wonder if the hydrophilicity of Cu-PCC is varied from the Cu-C counterpart due to its unique nanosized concave surfaces? If so, will the modulated hydrophilicity account for the enriched C₂H₂?

Answer 2: We acknowledge the reviewer’s helpful comments and suggestions. To eliminate the reviewer’s concerns, the contact angles of these two catalysts were measured to assess their hydrophilicity. As shown in Figure R2 (Supplementary Figure 16 in the revised Supplementary Information), there was no obvious difference in hydrophilicity between Cu-C and Cu-PCC. The corresponding revisions are displayed on a yellow background in the revised Manuscript as follows:

“The Raman, X-ray photoelectron spectroscopy (XPS), X-ray absorption spectroscopy (XAS), and contact angle results show that there are no other differences between Cu-PCC and Cu-C, other than the nanosized concave surfaces over the PCC supports.”

Figure R2 a, b) The contact angles of Cu-PCC (a) and Cu-C (b).

Comment 3. For performance evaluation, the reviewer noticed that there is a small fraction of C₂H₆ exist both in Cu-PCC and Cu-C, however, the signal of C₂H₆ in the online differential electrochemical mass spectrometry is absent without explanation.

Answer 3: We acknowledge the reviewer's constructive comments and suggestions. The disappearance of the C₂H₆ signal in the DEMS can be explained as follows: the detected C₂H₆ in the performance evaluation part is an accumulated result, and the signal for the online DEMS represents the transient status without accumulation. Since the production fraction of the C₂H₆ byproduct is quite low, the transient signal of DEMS is too weak to be detected.

Comment 4. The following is not criticism, just discussion:

Considering that the integrals of the IR bands could be related to the coverage of the respective adsorbate on the surface, using the area ratio to describe the relative coverage of two species is a quite insightful method, however, the recognition of specific species should be very careful. I have one technical question on how the author distinguish the characteristic peaks of H₂O and C₂H₂? And I think it is deserved to be written in the Supporting Information for the judgment of such a method.

Answer 4: We acknowledge the reviewer's kind and constructive comments and suggestions. First, we are grateful and happy for the recognition of our proposed method. It is indeed important for us to clarify the details of the peak distinguishing process, as the reviewer pointed out. Specifically, the isotope effect was analysed to determine the two characteristic peaks of H₂O and C₂H₂. In addition to the spectral comparison between C₂H₂ and the Ar atmosphere provided in the original version (Supplementary Figure 27 in the revised Supplementary Information). As shown in Figure R3 (Supplementary Figure 28 in the revised Supplementary Information), to eliminate the interference of the O-H bond in H₂O when identifying the location of C₂H₂*, D₂O was used to replace H₂O. The peaks located at approximately 3200 to 3300 cm⁻¹ and approximately 1600 cm⁻¹ could only be detected under a C₂H₂ atmosphere. Thus, these peaks were attributed to $\nu(\text{C-H})$ and $\nu(\text{C}\equiv\text{C})$, respectively. To save the reviewer's valuable time, key revisions are displayed on a yellow background in the revised Supplementary Information.

Figure R3 The results of the isotope effect after H₂O was replaced with D₂O under a C₂H₂ atmosphere.

To reviewer 2:

Reviewer letter: This manuscript proposes a design strategy using nanosized concave to regulate the interfacial microenvironment, which increase the local C₂H₂ concentration and optimize its mass transfer kinetics. Thus, the electrocatalytic semihydrogenation of raw coal-derived C₂H₂ performance over the as-designed Cu-PCC is greatly enhanced and enables a high C₂H₄ Faradaic efficiency and C₂H₂ single-pass conversion. In situ ATR-SEIRAS demonstrates the C₂H₂ molecule is enriched around the Cu-PCC interface and becomes easier to be activated, resulting in outstanding performance. In summary, this work is very interesting and coherent. Therefore, I recommend the acceptance of this manuscript after clarifying my concerns that listed below.

Answer: Thank you very much for your constructive suggestions to improve the quality of our work and for your positive comments on our revised manuscript. Regarding the concerns of the reviewer, we have provided a point-by-point response. To save the reviewer's valuable time, key revisions are displayed on a yellow background in the revised manuscript and Supplementary Information (SI). We are sure that the quality of this work will be greatly improved after being revised.

Comment 1. The author claimed that the adsorption of C₂H₂ on Cu is greater than that on carbon supports without specific comparison. That is there is no specific value of ICOHP of the acetylene molecule adsorbed over Cu in supplementary figure 1.

Answer 1: We acknowledge the reviewer's kind comments and suggestions. We apologize for the lack of a specific value for the ICOHP of the acetylene molecule. As shown in Figure R4, we have updated Supplementary Figure 1 in the revised Supplementary Information.

Figure R4 Projected crystal orbital Hamilton population ($-p\text{COHP}$) for the C-C interaction of gaseous C₂H₂.

Comment 2. The error bar in Fig. 5c is missing. It is suggested that the author supplement the experiment and improve credibility of Fig. 5c.

Answer 2: We acknowledge the reviewer's kind comments and suggestions. According to the reviewer's suggestion, an error bar has been added to Fig. 5c in the revised Manuscript (Figure R5) for its reliability.

Figure R5 The C₂H₄ yield normalized to the ECSA and Cu loading capacity.

Comment 3. The particle size of catalytic materials can significantly impact performance, and even induce different reaction mechanisms. The authors should provide statistical distributions of particle sizes for the two materials.

Answer 3: We acknowledge the reviewer's kind comments and suggestions. We agree with the reviewer's comments, and the particle size distributions (Figure R6 or Supplementary Figure 10 in the revised Supplementary Information) of Cu-PCC and Cu-C are provided in the revised Supplementary Information and Manuscript as follows:

"In addition, the size distributions are counted from three STEM images of Cu-PCC and Cu-C, respectively. As shown in supplementary figure x, the size of the majority of Cu particles in Cu-PCC is approximately 80 nm, which is slightly larger than that over Cu-C (~60 nm)."

"The better performance of Cu-PCC over its slightly larger particle size further demonstrates the superiority of the nanosized concave surface."

Figure R6 a, b) Statistical size distributions of Cu-PCC (a) and Cu-C (b).

Comment 4. In Figure 6c and d, the authors demonstrate slight differences in the Raman shift of acetylene on Cu-PCC and Cu-C. However, it should be careful when analyzing Raman signals in the range of 1000-1800 cm^{-1} due to that the skeletal signals of carbon materials also appear at this range. The authors should further clarify how they identify signals near 1700 cm^{-1} and ensure that they originate from the reactant acetylene rather than the reduced carbon carrier.

Answer 4: We acknowledge the reviewer's constructive comments and suggestions. We apologize for the reviewer's concerns due to our negligence. Indeed, we performed preexperiments for the identification of acetylene signals. As shown in Figure R7 (Supplementary Figure 32 in the revised Supplementary Information), the peak located in the range of 1650~1750 cm^{-1} corresponds to gaseous C_2H_2 at approximately 1800 to 2100 cm^{-1} under a C_2H_2 atmosphere, whereas no peak appears under an Ar atmosphere. The above results verify that the peak located at approximately 1650~1750 cm^{-1} is attributed to adsorbed C_2H_2 . These results have been added to the revised Supplementary Information.

Figure R7 Raman spectra of Cu-PCC and Cu-C at OCP under Ar and C_2H_2 atmospheres, respectively.

Comment 5. The description of IR and Raman characterization is too vague to be compared with other work. More details regarding signal acquisition should be provided for its reference value.

Answer 5: We acknowledge the reviewer's suggestion. Some necessary details about the IR and Raman characterization methods have been added to the Methods section of the revised manuscript as follows:

“Electrochemical in situ ATR-FTIR measurements. *In situ ATR-FTIR was performed on a Nicolet 6700 FTIR spectrometer equipped with an MCTA detector with silicon as the prismatic window and an ECIR-II cell by Linglu Instruments. First, Cu-PCC was carefully dropped on the surface of the gold film, which was chemically deposited on the surface of the silicon prismatic material before each experiment. Then, the deposited silicon prismatic material served as the working electrode. Pt foil and Hg/HgO with an internal reference electrolyte of 1.0 M KOH were used as the counter and reference electrodes, respectively. A 1 M KOH solution was used as the electrolyte. The electrolyte was presaturated with pure C₂H₂ gas, and the gas was continuously bubbled through during the whole measurement. Specifically, the isotope effect was determined to determine the location of the two characteristic peaks of H₂O and C₂H₂ (see details in the Supplementary Information). The spectrum was recorded every 30 s in the range of 1000-3600 cm⁻¹ under an applied potential ranging from 0.2 to -1.0 V vs. RHE.*

Electrochemical in situ Raman measurements. *In situ electrochemical Raman spectra were recorded via an electrochemical workstation on a Renishaw inVia reflex Raman microscope under 532 nm laser excitation under controlled potentials. We used a homemade Teflon electrolytic cell equipped with a piece of round quartz glass for laser application and protection of the tested samples¹. Before the experiments, the electrolyte was pretreated with pure C₂H₂ gas to obtain C₂H₂-saturated KOH. The working electrode was parallel to the quartz glass to maintain the plane of the sample perpendicular to the incident laser. The Pt wire was rolled to a circle around the working electrode to serve as the counter electrode. The reference electrode was Hg/HgO with an internal reference electrolyte of 1.0 M KOH. To confirm the location of the target peak of the C₂H₂ molecule, a series of controlled preexperiments were conducted under different atmospheres (see details in the Supplementary Information). The spectrum was recorded under applied potentials ranging from 0.2 to -1.0 V vs. RHE from 900 to 2200 cm⁻¹.”*

To reviewer 3:

Reviewer letter: Seeking a green and petroleum-independent alternative toward ethylene synthesis is significant to the manufacturing industry of commodity chemicals, medicines, and agrochemicals. This manuscript demonstrated the electrocatalytic hydrogenation of acetylene derived from coal produce ethylene and focused on directly using the low concentration raw coal-derived C₂H₂ as feedstocks for further decrease the costs of C₂H₄ production. They proposed and confirmed that the negative polarization field caused by the concave surface enhances the conjugate action between C₂H₂ and the carbon support, which is suitable for C₂H₂ enrichment. Besides, the unique concave support could also boost the mass transfer kinetics of C₂H₂ toward Cu sites. This interesting work is an impressive breakthrough in the petroleum-independent C₂H₄ electrosynthesis, thus I would like to recommend this interesting work for being accepted after addressing the following minor issues:

Answer: Thank you very much for your kind comments and constructive suggestions on our manuscript. Regarding the concerns of the reviewer, we have provided a point-by-point response. To save the reviewer's valuable time, key revisions are displayed on a yellow background in the revised manuscript and Supplementary Information (SI). We are sure that the quality of this work will be greatly improved after being revised.

Comment 1. The author proposed a sequential self-template transformation method based on the Kirkendall effect to construct Cu-PCC. However, the details of TA exchange process is missing.

Answer 1: We acknowledge the reviewer's helpful comments and suggestions. We were negligent in losing the details of the TA exchange process. The optimization parameters of the transformation process are important for the generalizability of our work. Thus, SEM images (Figure R8 or Supplementary Figure 6 in the revised Supplementary Information) of Cu-TA with different TA contents and exchange times have been added to the revised Supplementary Information.

"The optimization process is shown in Supplementary Figure x. As a result, the Cu-MOF precursor could not be etched under a low TA concentration and short exchange time, while the parent octahedral morphology would be broken under a high TA concentration and long exchange time. Thus, 1 mg/ml TA solution and a 7 h exchange time were chosen as the optimum conditions."

Figure R8 Optimization process of the TA exchange process. (a-c) Cu-TAs obtained using 0.5, 1.0, and 2.0 mg/ml TA solutions, respectively. (d-f) Cu-TAs obtained after etching times of 5, 7, and 9 h, respectively.

Comment 2. As for the X-ray absorption spectroscopy results in Supplementary Figures 11b, c, it is meaningless to provide only the comparison of Cu-PCC and Cu-C without the standard results of Cu foil.

Answer 2: We acknowledge the reviewer's constructive comments and suggestions. To improve the reliability of our XAFS results, we have added the standard results of Cu foil to the revised Supplementary Information (Figure R9 or Supplementary Figure 15b, c in the revised Supplementary Information) for the strictness of the proposed data.

Figure R9 a, b) X-ray absorption near-edge structure spectra (a) and EXAFS (b) spectra of Cu-PCC and Cu-C.

Comment 3. The whole manuscript focused on the modulation of C₂H₂ feedstock, however, H₂O, as a hydrogen source, should also be considered. As the recently reported electrocatalytic hydrogenation cases, the kinetic isotope effect experiment should be carried out to supplement the reaction mechanism.

Answer 3: We acknowledge the reviewer's helpful comments and suggestions. To further understand the reaction mechanism of the ESAE process using H₂O as a hydrogen source, kinetic isotope (KIE) experiments were performed within the same potential range for performance evaluation and *in situ* characterization. As shown in Figure R10 (Supplementary Figure 26 in the revised Supplementary Information), the k_H/k_D values among the four potentials selected over Cu-PCC and Cu-C are all greater than 2, indicating that the hydrogen-reliance nature of the ESAE process has not changed over the proposed Cu-PCC. To save the reviewer's valuable time, the corresponding revisions are displayed on a yellow background in the revised Manuscript and Supplementary Information and Manuscript as follows:

"First, the kinetic isotope effect (KIE) measurements show that the k_H/k_D values among the potentials selected over Cu-PCC and Cu-C are all in the range of 1~2 with a similar trend (Supplementary Fig. X), indicating that the hydrogen-reliance nature of the ESAE process has not changed due to the concave support."

Figure R10 a, b) The KIEs of Cu-PCC (a) and Cu-C (b) under different applied potentials.

Comment 4. Additionally, I have noticed that the authors claim that their products were quantified using GC-2010 gas chromatography, but I did not find any information of the method they applied. Is that the calibration curve method? Or internal standard method? The author should pay much attention to such details.

Answer 4: We acknowledge the reviewer’s constructive comments and suggestions. Indeed, the calibration curve method was applied for our quantification, and we apologize for the lack of information about the calibration curves used in our work. The corresponding quantitative process in the Methods section has been updated in the revised Manuscript. As shown in Figure R11 (Supplementary Figure 21 in the revised Supplementary Information), the calibration curves have been added to the revised Supplementary Information in the present version. We have rechecked the entire manuscript to avoid such mistakes. To save the reviewer’s valuable time, key revisions are displayed on a yellow background in the revised Supplementary Information.

“Quantitative analysis of the C₂H₂ conversion, evolution rate, and FE of the obtained products. The products were subjected to a GC-2010 gas chromatograph equipped with an activated carbon-packed column (with He as the carrier gas) and a barrier discharge ionization detector. The C₂H₂ conversion and evolution rate of the different products were calculated using equations (1)-(4), and the FEs of the different products were calculated using equation (5). All the experiments were repeated three times.

$$C(X)=k(X)\times\text{peak area}$$

(1)

$$n(X)=C(X)\times V$$

(2)

$$\text{Conversion (\%)} = \frac{n(\text{C}_2\text{H}_2)_{\text{in}} - n(\text{C}_2\text{H}_2)_{\text{out}}}{n(\text{C}_2\text{H}_2)_{\text{in}}} \times 100\% \quad (3)$$

$$\text{Evolution Rate (mmol/mg/h)} = \frac{n(X)}{m} \times S \quad (4)$$

$$FE_X(\%) = \frac{a \times n_X \times F}{Q}$$

(5)

X: The feedstock and products, including C₂H₂, H₂, C₂H₄, and C₂H₆.

C: The concentration of feedstock and products.

m: The mass of catalysts over the electrode.

n: The moles of feedstock and products.

k: The slope of the calibration curves for feedstock and products.

S: The gas flow rate.

a: The electron transfer number.

F: Faraday constant.

Q: The total Coulomb number of the ESAE process.”

Figure R11 a-d) The corresponding standard curves of H₂ (a), C₂H₄ (b), C₂H₆ (c), and C₂H₂ (d) with different volume fractions for quantitation.

Comment 5. There are also some format and images display issues in the present manuscript which should be addressed, such as Figures 4b, c, and d (there are strange lines exist), Supplementary Figure 8c.

Answer 5: We acknowledge the reviewer's helpful comments and suggestions. This is our mistake for the ill display of figures in our original version. We have revised and rechecked the entire manuscript and Supplementary Information to avoid such mistakes and improve the readability of our work.

Comment 6. The performance of C₂H₄ production from the electrocatalytic semihydrogenation of C₂H₂ in this work should be compared with that of recently reported catalysts to determine its significance and innovation.

Answer 6: We acknowledge the reviewer's helpful comments and suggestions. According to the reviewer's constructive suggestion, a comparison of the performance of the Cu-PCC and the

state-of-the-art reports on ESAE was added to the revised Supplementary Information (Table R1 or Supplementary Table 1 in the revised Supplementary Information).

Table R1 Comparison of the performance of the Cu-PCC and the state-of-the-art reports on the ESAE.

Catalysis	$C_{C_2H_2}$ / %	Electrolyte	$j_{C_2H_4}$ mA cm ⁻²	Potential	FE _{C₂H₄} / %	Specific selectivity / %	Reference
Cu-PCC	15	1 M KOH	-420	-1.2 V vs. RHE	91.53	100	This work
Ag NWs	1	1 M KOH	-2.2	/	/	100	CCS Chem. 2023, 5, 200-208
LD-Cu	5	1 M KOH	-61.9	-0.6 V vs. RHE	74.9	~92	Nat. Catal. 2021, 4, 565-574
ED-Cu NPs	100	1 M KOH	-488.7	-1.93 V vs. RHE	97.7	100	Nat. Sustain. 2023, 6, 827-837
Cu-MPs	/	1 M KOH	~-26	-0.9 V vs. RHE	~40	~80	Nat. Commun. 2021, 12, 7072
Cu dendrites	100	1 M KOH	-150	-0.8 V vs. RHE	~93	~98	Nat. Catal. 2021, 4, 557-564
NHC-Cu	100	1 M KOH	-158.8	-0.9 V vs. RHE	98	96	Nat. Commun. 2021, 12, 6574
2TIm		1 M KOH	-225	-0.9 V vs. RHE	98	98	Nat. Chem. 2024, 10.1038/s41557- 024-01480-6

Comment 7. The characterization of the Cu-PCC after electrocatalytic stability should be provided to support its long-term stability and confirm its true active sites.

Answer 7: We acknowledge the reviewer's helpful comments and suggestions. According to the reviewer's constructive suggestion. Although the SEM, HRTEM and XRD results after the stability test have already been provided in the original version (Supplementary Figure x in the revised Supplementary Information), Cu 2p XPS was conducted to further eliminate the reviewer's concerns about the long-term stability and the true active sites. As shown in Figure R12 (Supplementary Figure 25d in the revised Supplementary Information), the binding energy of Cu remains unchanged after the stability test, indicating that the metallic Cu site is the true active site of the ESAE process.

Figure R12 Cu 2p XPS spectra after the stability test.

To reviewer 4:

Reviewer letter: In this manuscript, the authors focus on the raw coal-derived C₂H₂ semi-hydrogenation process and design a porous concave carbon-supported Cu nanoparticle (Cu-PCC) restraining the competing HER. The well-designed carbon carrier with nanosized concave is synthesized through a facile self-template and benefits for the mass transform and the activation of C₂H₂. Using a low concentration of C₂H₂, the Cu-PCC deliver a high C₂H₄ Faradaic efficiency and a good single-pass C₂H₂ conversion at -1.2 V vs RHE, outperforming the counterpart without concave surface. The origin of performance enhancement is also clarified. I believe that this work is highly recommended for its publication in Nature Communications after minor modifications.

Answer: Thank you very much for your kind comments and constructive suggestions on our manuscript. Regarding the concerns of the reviewer, we have provided a point-by-point response. To save the reviewer's valuable time, key revisions are displayed on a yellow background in the revised manuscript and Supplementary Information (SI). We are sure that the quality of this work will be greatly improved after being revised.

Comment 1. Parameters used in the molecular dynamics simulation calculations should be explicitly stated as they can significantly influence the simulation results.

Answer 1: We acknowledge the reviewer's helpful comments and suggestions. As the reviewer says, the parameters do matter the simulation results; thus, the computational model has been provided in the "Computational details" section in the revised Manuscript, which is extracted below:

"In the ab initio molecular dynamics (AIMD) simulations, canonical ensemble (NVT) conditions were imposed by a Nose-Hoover thermostat with a target temperature of 300 K. The MD time step was 1 fs, and all the systems were run for 10 ps to reach equilibrium. In the plane and angle models, 114 and 120 water molecules were added to ensure that the density of water in the model was approximately 1 g/cm³. The last 1 ps of data in the AIMD process are selected for analysis. In the process of hydrogen bond analysis, we set the maximum distance of the hydrogen bond to 3.5 Å and the angle cut-off to 40°.

Comment 2. The comparison of the distance between water clusters and the catalysts layer in Figure 2d is intuitive, yet the authors have not conducted further quantitative analysis. Supplementing this discussion would make the conclusion more comprehensive.

Answer 2: We acknowledge the reviewer's constructive comments and suggestions. We apologize for the lack of quantitative analysis of the distance between the water clusters and catalyst layer. To eliminate the reviewer's concerns, the radial distribution functions (RDFs) between the C-H bonds, which are regarded as descriptors of the distance between the water cluster and the carbon layer, were

calculated (Figure R13) and added to the revised Supplementary Information. The corresponding revision in the revised Manuscript is as follows:

“To further quantify d over the C and PCC models, the radial distribution functions (RDFs) between C-H were calculated. As shown in Supplementary Fig. X, $g(r)_{C-H}$, which is closely associated with d , shows an average increase over the PCC model at approximately 0.1 Å compared to its C counterpart. This result indicates that the d over the concave C layer is lengthened.”

Figure R13 The calculated RDF of C-H in the PCC and C.

Comment 3. The author did not include cross-sectional data in Figure 3f. It is suggested that the author supplement these data and the raw format of AFM data.

Answer 3: We acknowledge the reviewer’s kind comments and suggestions. To strengthen the reliability of our results, the original AFM images (Figure R14a, b or Supplementary Figure 11a, b in the revised Supplementary Information) have been added to the revised Supplementary Information. In addition, the surface height distributions of the two catalysts were also collected to further verify the existence of the nanosized concave surface (Figure R14c, d or Supplementary Figure 11c, d in the revised Supplementary Information).

Figure R14 a, b) The original AFM images of Cu-PCC (a) and Cu-C (b); c, d) The heights distribution of the cross profile between the lowest and highest point.

Comment 4. Cu Auger LMM spectra, which could provide more sensitive surface valence state information compared to Cu 2p spectra, should be added to eliminate the activity contribution from electronic structures differences.

Answer 4: We acknowledge the reviewer's comments and suggestions. For the electronic differences between Cu sites in Cu-PCC and Cu-C, we have already provided several experimental results (Supplementary Figures 15a-c in the revised Supplementary Information), such as Cu 2p XPS, XANES and EXAFS, to demonstrate that the valence state and coordinate environment of Cu are almost the same between Cu-PCC and Cu-C. However, to further eliminate the reviewer's concerns, Cu Auger LMM spectra were obtained to explore the surface valence state. As shown in Figure R15 (Supplementary Figure 15d in the revised Supplementary Information), in addition to the Cu²⁺ species caused by inevitable oxidation under air conditions, Cu⁰ and Cu⁺ are both detected over Cu-PCC and Cu-C, further indicating that the enhanced activity is independent of the differences in the electronic structure.

Figure R15 The Cu Auger LMM spectra of Cu-PCC and Cu-C.

Comment 5. There are some minor formatting and spelling issues in the text, such as in Fig. S25a, where the x-axis of "energy change profiles" should be "reaction coordinate"; the significant figure precision of ICOHP in Fig. 2b. Additionally, there are instances of minus and hyphen misuses. The author should carefully review the manuscript.

Answer 5: We acknowledge the reviewer's kind comments and suggestions. We apologize for our negligence in these formatting and spelling mistakes in our original version. We have revised and rechecked the entire manuscript and Supplementary Information to avoid such mistakes and improve the readability of our work.

We thank all the four reviewers for their kind and professional suggestions.

We are sure that the quality of this work will be greatly improved according to these helpful comments and suggestions from the four reviewers.

REVIEWERS' COMMENTS

Reviewer #1 (Remarks to the Author):

The authors have undertaken considerable efforts to strengthen the claims presented in the manuscript and I can now recommend this manuscript for publication in Nat. Commun.

Reviewer #2 (Remarks to the Author):

The authors have amended manuscript according to the comments, therefore, I recommend to accept this work at present.

Reviewer #3 (Remarks to the Author):

The authors have properly addressed previous concerns. The current version can be published.

Reviewer #4 (Remarks to the Author):

The authors has well solved the problem I put forward. I suggest that the journal accept the paper.

A point-by-point response to the reviewers

To reviewer 1:

Reviewer letter: The authors have undertaken considerable efforts to strengthen the claims presented in the manuscript and I can now recommend this manuscript for publication in Nat. Commun.

Answer: We highly appreciate the reviewer's very positive comments on our work. We are sure that the quality of this work has been greatly improved according to these comments and suggestions.

To reviewer 2:

Reviewer letter: The authors have amended manuscript according to the comments, therefore, I recommend to accept this work at present.

Answer: We highly appreciate the reviewer for his/her positive comments on our revised manuscript. We are sure that the quality of this work has been greatly improved according to these comments.

To reviewer 3:

Reviewer letter: The authors have properly addressed previous concerns. The current version can be published.

Answer: We highly appreciate the reviewer's positive comments on our revised manuscript. We are sure that the quality of this work has been greatly improved according to these comments.

To reviewer 4:

Reviewer letter: The authors has well solved the problem I put forward. I suggest that the journal accept the paper.

Answer: We highly appreciate the reviewer's positive comments on our revised manuscript. We are sure that the quality of this work has been greatly improved according to these comments.